# Effect of Dietary Fiber (Oat Bran) Supplement in Heart Rate Lowering in Patients with Hypertension: A Randomized DASH-Diet-Controlled Clinical Trial

**DOI:** 10.3390/nu14153148

**Published:** 2022-07-30

**Authors:** Yang Ju, Chenglin Zhang, Zhirong Zhang, Hongying Zhu, Yuanyuan Liu, Ting Liu, Omorogieva Ojo, Jingbo Qiu, Xiaohua Wang

**Affiliations:** 1Nursing Department, The First Affiliated Hospital of Soochow University, Suzhou 215006, China; juyang060509@163.com (Y.J.); 20195231002@stu.suda.edu.cn (Y.L.); lt322426@163.com (T.L.); 2Nursing Department, The Yancheng School of Clinical Medicine, Nanjing Medical University, Yancheng 224000, China; chenglin_zcl@163.com; 3Division of Cardiology, The General Public Hospital of Zhangjiagang, Zhangjiagang 215699, China; zyy9812@163.com (Z.Z.); qlkzhy@163.com (H.Z.); 4School of Health Sciences, Faculty of Education, Health and Human Sciences, University of Greenwich, London SE9 2UG, UK; o.ojo@greenwich.ac.uk; 5Nursing Department, The International Peace Maternity and Child Health Hospital, School of Medicine, Shanghai Jiao Tong University, Shanghai 200030, China; ilmen@163.com; 6Division of Cardiology, The First Affiliated Hospital of Soochow University, Suzhou 215006, China

**Keywords:** hypertension, dietary fiber, heart rate

## Abstract

(1) Background: The management goal for patients with essential hypertension (HTN) is not only to lower blood pressure (BP), but also to control increased heart rate (HR). In a previous study, it was found that dietary fiber (DF) supplementation can effectively reduce BP in patients with HTN. The aim of this study was to determine whether a DF supplement can lower HR in patients with HTN. (2) Methods: Seventy patients who met the inclusion and exclusion criteria were randomly allocated into the control group (*n* = 34) and the intervention group (*n* = 36). The regular DASH dietary care was delivered to both groups of patients. In addition, one bag of oat bran (30 g/d, containing DF 8.9 g) was delivered to the intervention group. The 24 h ambulatory heart rate was measured at baseline and 3 months. (3) Results: At 3 months, the 24 h maximum heart rate (24h maxHR) in the intervention group was significantly lower than that in the control group. After the intervention, within-group comparisons in the intervention group revealed that there were significant reductions in the 24 h average heart rate (24h aveHR), 24h maxHR, average heart rate during day time (D-aveHR), minimum heart rate during day time (D-minHR), and maximum heart rate during day time (D-maxHR). Similar differences were not found in the control group. (4) Conclusions: Dietary fiber (oat bran) supplementation might be beneficial in lowering HR in patients with HTN.

## 1. Introduction

The prevalence of hypertension (HTN) is on the increase globally [1]. In 2000, the worldwide number of adult patients with HTN was reported to be 972 million, while the number is estimated to reach 1.56 billion by 2025 [2]. In China, a nationwide survey reported that 23.2% of the population whose age was 18 years or older had HTN [2]. HTN is one of the major risk factors that contribute to cardiovascular and kidney diseases, also being a leading cause of premature mortality [3]. The poor prognosis of hypertensive patients is not only related to poor-controlled blood pressure, but also closely related to an increased heart rate [4]. An increased heart rate is associated with new heart failure [5] and all-cause and cardiovascular deaths [6]. Saxena reported that an increase in basic heart rate of 10 bpm was associated with a 22% increase in the risk of heart disease or death in patients with HTN [7]. A cohort analysis of patients followed up with refractory hypertension reported the lowest incidences of fatal and non-fatal cardiac events and all-cause deaths at heart rates of 60–75 bpm, while all-cause and cardiovascular deaths increased at heart rates of >75 bpm [6]. Therefore, in addition to blood pressure control, attention should be paid to HR management for hypertensive patients as recommended by the Chinese Expert Consensus [8]. In fact, management of heart rate in patients with hypertension is also supported by the 2018 European Society of Cardiology/European Society of Hypertension (ESC/ESH) guidelines for management of hypertension [9].

However, the current status of HR management in HTN patients is not optimistic. A cross-sectional survey of 115,229 patients with hypertension in 21 cities in China showed that the resting HR of over 75 bpm accounted for 55.9% [5]. Beta-blockers are the preferred drug to control heart rate in patients with high blood pressure [10]. Sun et al. [5] conducted a survey and found that only 36.9% of HTN patients used beta-blockers, and the average heart rate of hypertensive patients in China was 76.6 bpm, which was higher than the recommended standard of 70 bpm by the Consensus of Asian Experts [11].

There is evidence that non-pharmacological therapies such as breathing exercises and yoga could improve heart rate [12,13]. For example, Zou et al. conducted a meta-analysis and found that the breathing exercise in patients with cardiovascular diseases resulted in significant reduction in HR (mean difference: −1.72 bpm, 95% CI −2.70 to −0.75) [13]. Moreover, 12 weeks of yoga has been shown to produce a significant reduction in HR of patients with paroxysmal atrial fibrillation [12]. Dietary nutrition therapy, as a non-drug treatment, such as suggesting improvement of body weight, quality of diet, and eventually inclusion of some dietary supplements [14], plays a vital role in the management of HTN [15,16]. Current dietary recommendations for patients with hypertension include increasing fruit, vegetable, and nut consumption and adherence to Dietary Approaches to Stopping Hypertension (DASH)-type dietary patterns and the Mediterranean diet plan [17]. All these dietary recommendations include diets that are rich in dietary fiber (DF).

DF is called the “seventh nutrient” and it is the fraction of the edible part of plants that are resistant to digestion and absorption in the human small intestine [18,19]. It is an important component of the recommended vaso-protective dietary approaches [18]. DF intake has been reported to be beneficial in reducing both BP and serum cholesterol, and so it is believed that a deficiency in DF might be contributing to the epidemic of cardiovascular disease [20]. A comprehensive meta-analysis of 58 clinical trials and 3974 subjects concluded that the inclusion of oat-containing foods in the diet may be a valid strategy to prevent the onset of cardiovascular disease [21]. Another meta-analysis of 25 randomized controlled trials revealed that DF supplement was related to a significant decrease in mean systolic blood pressure (SBP) by 5.95 mmHg (95% CI, −9.50 to −2.40) and diastolic blood pressure (DBP) by 4.20 mmHg (95% CI, −6.55 to −1.85) compared with the omnivorous diets [22]. A study by our team [23] indicated that using oat bran 30 g/d supplement improved 24 h maximum and average SBP (14.0 ± 15.5 mmHg and 4.5 ± 8.1 mmHg, respectively); 24 h maximum and average DBP (11.1 ± 14.6 mmHg and 3.1 ± 5.6 mmHg, respectively); and also modulated the gut microbiota, potentially contributing to the regulation of autonomic nerves and the renin–angiotensin–aldosterone system [24], thus potentially reducing HR. On the basis of the information above, we speculated that an increased dietary fiber intake should also be beneficial in reducing heart rate of HTN patients. However, there are few studies that have focused on the relationship between DF and HR in HTN patients. We hypothesized that in addition to improving blood pressure, dietary fiber supplementation can also reduce heart rate in patients with HTN. In order to test this hypothesis, HTN patients were recruited into this randomized controlled trial, and heart rates were measured and analyzed.

## 2. Materials and Methods

### 2.1. Study Design and Population

This study was a randomized DASH-diet-controlled trial (RCT) that was performed from March 2019 to September 2020. All subjects gave their informed consent for inclusion before they participated in the study. The study was conducted in accordance with the Declaration of Helsinki, and the protocol was approved by the Ethics Committee of the Soochow University (project identification code: ECSU-2019000148). The clinical trial registration number for this study is ChiCTR2000033987. All participants signed a written informed consent form. Eligible participants were randomly and blindly allocated to the intervention (dietary fiber, DF) group or the control group using computer-generated random numbers and accepted the 3 months intervention period. Before the intervention, all participants had a 1-week washout period to reduce the effect of background diets on this study. Participants were blinded after assignment to intervention.

### 2.2. Subjects

Participants with hypertension were enrolled from Bai-Liang community and outpatient department of the No. 1 Affiliated Hospital of Soochow University in Suzhou, China. The inclusion criteria were as follows: (1) 18 to 65 years of age and diagnosed with hypertension stage 1, according to the latest Guidelines for the Prevention and Treatment of Hypertension in China [25]; (2) without change of antihypertensive drugs for two weeks before intervention; (3) able to communicate; (4) willingly to be involved in this study and signed the informed consent form. The exclusion criteria were as follows: (1) allergic to oat bran or being enrolled in other studies; (2) regularly eating dietary fiber ≥ 25 g/d; (3) with complications and serious diseases (e.g., renal failure or shock); (4) having gastrointestinal diseases in the past month; (5) taking antibiotics, hormones, or microecological agents in the past month; (6) lactating or pregnant woman; and (7) having a cognitive impairment or serious mental disorder. Participants whose oat bran intake < 4 d/w during intervention in the intervention group were excluded.

### 2.3. Sample Size Calculation

We did not find a similar study of exploring the effect of DF-supplement on lowering heart rate in patients with hypertension. Before commencing the research, we designed a pre-experiment with 10 participants and calculated the sample size on the basis of the results of the pre-experiment. The mean difference and standard deviation (SD) of office heart rate at the end of the intervention between the two groups were 4 and 5 bpm. With α = 0.05, power = 0.85, we calculated 28 patients for each group. In view of the sample loss of 20%, the number for each group was 34. Finally, we recruited 35 patients for each group in the study.

### 2.4. Intervention

#### 2.4.1. Control Group

According to The Guidelines for the Prevention and Treatment of Hypertension in China (2018 Revised Edition), a usual dietary program education (Dietary Approaches to Stop Hypertension Diet, DASH) for patients with hypertension [26] was recommended to the control group at participants’ first visit. The DASH dietary pattern is rich in fruit, vegetables, whole grains, and low-fat dairy products, whereas the saturated and total fats were quite low [27].

#### 2.4.2. Intervention Group

On the basis of the dietary education of the control group, patients in the intervention group were delivered oat bran of 30 g/d (DF content: 8.9 g/d with a vacuum packing). The oat bran was provided by Fuzhiyuan company, Xinji, China. The main researcher informed the patient to consume oat bran with breakfast or between meals and record the oat bran intake daily. If there were any adverse reactions, the participant was advised to inform the researchers immediately.

### 2.5. Follow-Up

Using face-to-face communication, WeChat, and cellphone, a 3-month follow-up was conducted in this study. The frequency was as follows: once a week in the first month, once every two weeks in the remaining two months. During the follow-up, the researcher assessed the patients’ diet, BP, exercise, and medication change. For the intervention group, the number of bags (30 g/bag) of oat bran consumed each week and side effects of oat bran were collected. Participants whose diet did not meet the requirements of the dietary program (oat bran intake < 4 bags per week) during intervention were excluded from the study.

### 2.6. Outcomes

#### 2.6.1. Primary Outcomes

The primary outcomes were 24 h average heart rate (24h aveHR), 24 h minimum heart rate (24hminHR), and 24 h maximum heart rate (24h maxHR); average heart rate during day time (D-aveHR), minimum heart rate during day time (D-minHR), and maximum heart rate during day time (D-maxHR); average heart rate during the night (N-aveHR), minimum heart rate during the night (N-minHR), and maximum heart rate during the night (N-maxHR). The secondary outcomes were the changes of medications affecting heart rate.

#### 2.6.2. Secondary Outcomes

##### Compliance to Oat Consumption

On the basis of the collected number of bags of oat bran consumed every week, the compliance to oat bran consumption was evaluated.

##### Side Effect of Dietary Fiber

During follow-up, we collected the side effects due to consumption of oat, treatment, and relief time.

##### Change of Medication

We collected data on the number of doses that were increased or unchanged, as well as reductions of beta blockers, ivabradine, and calcium ion antagonist that the participants took during the intervention.

Increased: during intervention, an increase in the dose of the drugs compared to the baseline or from not taking the drugs to taking the drugs; no change: there was no dose adjusted for drugs during intervention; reduced: during intervention, a reduction in the dose of the drugs compared to the baseline or from taking the drugs to not taking the drugs.

### 2.7. Measurement

#### 2.7.1. Clinical Data and Anthropometric Measurements

A set of general questionnaires that included age, sex, marital status, educational level, occupational status, medical payment, exercise, duration of sleep, quality of sleep, smoking status, alcohol consumption, body mass index (BMI), waist-to-hip ratio (WHR), constipation, duration of hypertension, taking anti-hypertensive drugs, complications, and comorbidity were administered and assessed via a survey. Body height and weight were measured to the nearest 0.1 cm and 0.1 kg, respectively, in a standardized procedure with participants wearing light clothes and without shoes. BMI was calculated as weight (kg) divided by the square of height (m^2^). A tape was used to measure their waist and hip circumferences at the hospital or community. Waist–hip ratio was calculated as waist circumference divided by hip circumference. The quality of sleep was scored by a Visual Analog Scale according to participant’s perception. The scores less than 3 were classified as poor sleep quality, 4–6 as fair sleep quality, and more than 7 as good sleep quality [28]. Constipation was assessed with the validated Rome III criteria (straining (≥25%); lumpy or hard stools (≥25%); sensation of incomplete evacuation (≥25%); sensation of anorectal obstruction/blockade (≥25%); manual maneuvers to facilitate defecations (≥25%) and <3 times/w of defecations) and diagnosed in patients who met at least 2 of the 6 criteria [29]. The exercise referred to engaging in exercises other than daily activities. Exercise less than 3 times a week, less than 20 min each time, and continuous time less than 3 months was considered irregular exercise; otherwise, it is exercise [30]. Hypertension complication referred to the occurrence or development of another disease or symptom caused by blood pressure, including stroke, cerebral hemorrhage, myocardial infarction, frequent angina, aortic dissection, renal insufficiency, heart failure, and left ventricular hypertrophy [31]. They were selected via tick boxes by patients. Comorbidity referred to the co-existence of one or more diseases or clinical conditions and is independent of blood pressure [32]. When the patient cannot determine whether his disease was a complication, the research team asked the patients of medical history to make a decision.

#### 2.7.2. Diet and Oat Bran Consumption Record

A three-day diet diary was used to assess the quantity of participants’ DF and other nutrient intake. At baseline and 3 months, the researchers educated participants to record 3 days (2 working days and 1 weekend day) of detailed dietary consumption at home. The Chinese CDC nutrition calculator (Fei Hua nutrition software, Beijing, China) was used to calculate the quantity of each nutrient. Participants were instructed to record the number of bags of oat bran intake (30 g/bag) every week.

#### 2.7.3. The 24 h Heart Rate

Using an ABP device (Mobil-O-Graph PWA, I.E.M. Industrielle Entwicklung Medizintechnik, Stolberg, Germany), which can not only take BP measurements but also HR measurements, the data of 24 h HR was collected. HR was measured once every 30 min at night from 10:00 p.m. to 8:00 a.m. and every 20 min during the day time from 8:00 a.m. to 10:00 p.m. [33]. The criteria for valid HR were based on the criteria for valid BP (at least one BP measurement per hour successful recording of ≥80% of SBP and DBP during both the daytime and nocturnal periods), as well as valid range of HR from 20 to 250 bpm [34]. According to the above standard, when summarizing the 24 h BP and HR report, the data obtained should be strictly screened and checked. A total of 8 invalid blood pressure measures and 7 invalid heart rate measures were found and excluded in this study.

### 2.8. Statistical Analysis

Statistical analyses were performed using SPSS 19.0 software (SPSS, Inc., Chicago, IL, USA).

(1)Demographic, clinical data, and nutrients: First of all, the Kolmogorov–Smirov test and the Levene’s test were applied to test whether the data of different variables were normally distributed and of equal variance, respectively. The continuous variables being normally distributed were expressed as mean (M) ± standard deviation (SD), and comparisons between groups were made using the independent samples *t*-test if the variables were normally distributed and of equal variance; otherwise, the data of variables were expressed as median (P25, P75), and their comparisons between groups were made using the Mann–Whitney U test. For categorical variables, the data were described as the frequency (percentage), and comparisons between the two groups was conducted using the Pearson chi-squared test or Fisher’s exact test.(2)Comparisons of HR between the two groups were conducted at baseline and 3 months: if it was normally distributed, the comparison between two groups was carried out using the independent samples *t*-test; otherwise, the Mann–Whitney U test was used. For the changes of medication, Pearson’s chi-squared test or Fisher’s exact test was used for comparison. With respect to demographic data, clinical data, and nutrient data, analysis of covariance was applied to correct the relative biases if the comparison of variables between two groups was *p* < 0.1.(3)The paired *t*-test was used to compare the changes of HR within a group before and after intervention.(4)Intention-to-treat of HR was performed to ensure the reliability of results. A *p*-value of <0.05 was considered statistically significant.

## 3. Results

### 3.1. Baseline Characteristics

A total of 125 patients were recruited into this study, of whom 81 met the inclusion and exclusion criteria. During washout period, 2 lost contact and 4 lived far away from the hospital, leaving 75 participants who were left to sign the informed consent form. There were five patients who refused to sign the informed consent form. Among the remaining 70 patients, 36 were randomized to the intervention group and 34 to the control group. Three participants in the intervention group and four in the control group withdrew during the study. Finally, there were 33 participants in the intervention group and 30 in the control group (Figure 1). The patients in the intervention group were mainly male (78.8% vs. 53.3%), and the differences of baseline characteristics in other variables were not significant between the two groups (Table 1).

On the basis of the analysis of the food diary, the nutrients consumed at baseline were compared between the two groups. There were no significant differences in daily energy and nutrient (except DF) intake between the two groups at baseline (*p* > 0.05), which is shown in Table 2.

### 3.2. Dietary Fiber Intake

The quantity of dietary fiber intake was evaluated according to the frequency of oat bran consumption and the 3-day food diary. The result indicated that there was no significant difference in the quantity of DF intake between the two groups at baseline, while the quantity of DF in the invention group at 3 months was significantly more than that in the control group (21.45 ± 3.86 g/d vs. 10.40 ± 3.76 g/d, *p* < 0.001, Table 3). In addition, there was an average of 6.2 bags/w of oat bran consumption.

### 3.3. Effect of Dietary Fiber (Oat Bran) Supplementation on HR in Patients with Hypertension

All of the values of HR are shown in Table 4. At baseline, there were no statistically significant differences in all heart rate measures between the intervention group and the control group.

#### 3.3.1. 24h aveHR, 24hminHR, and 24h maxHR

At 3 months, the 24h maxHR in the intervention group was lower than that in the control group ((93.24 ± 15.26 vs. 98.17 ± 16.99) bpm, *p* = 0.049). On the basis of within-group comparisons after intervention, the results in the intervention group revealed there were significant reductions in the 24h aveHR ((75.15 ± 9.69 vs. 71.03 ± 9.91) bpm, *p* = 0.022) and 24h maxHR ((103.03 ± 18.43 vs. 93.24 ± 15.26) bpm, *p* = 0.006), while similar differences were not found in the control group. The results of intention-to-treat (ITT) related to 24h aveHR and 24h maxHR are shown in Table 5. The results showed the changes of HR were consistent with the findings in Table 4.

#### 3.3.2. D-aveHR, D-minHR, and D-maxHR

At 3 months, within-group comparisons in the intervention group revealed that there were significant reductions in the D-aveHR ((78.30 ± 10.53 vs. 73.30 ± 10.13) bpm, *p* = 0.006), D-minHR ((60.61 ± 9.05 vs. 56.97 ± 8.01) bpm, *p* = 0.049), and D-maxHR ((101.76 ± 20.73 vs. 92.79 ± 15.26) bpm, *p* = 0.030). However, similar differences were not found in the control group. Although there was no significant difference in the D-aveHR between the two groups, downward trends were found in D-aveHR ((5.00± 9.71 vs. 1.46 ± 6.48) bpm, *p* = 0.098) and D-maxHR ((8.97 ± 22.63 vs. 0.67 ±13.86) bpm, *p* = 0.088). The results of intention-to-treat (ITT) related to D-aveHR, D-minHR, and D-maxHR are shown in Table 5. The results showed that the changes of related HR were consistent with the findings in Table 4.

#### 3.3.3. N-aveHR, N-minHR, and N-maxHR

In this study, we did not find any statistical changes in the N-aveHR, N-minHR, and N-maxHR between the two groups and within-groups (Table 4 and Table 5).

### 3.4. Change of Medication

We compared the change of beta-blockers and calcium ion antagonist that the participants took after intervention, and the results showed that there was no significant difference in the use of HR-lowering drugs between the two groups (Table 6). During the trial period, no participants in either group used ivabradine.

### 3.5. Side Effect of Dietary Fiber

During follow-up, we did not find any side effects or symptoms due to oat consumption among the participants.

## 4. Discussion

An increased heart rate can be an independent cardiovascular risk factor [35], while dietary fiber is reportedly inversely associated with the risk of cardiovascular disease [36] and mortality from cardiovascular disease [37]. In this study, we explored the association between dietary fiber and heart rate, and the results confirmed the beneficial effect of dietary fiber supplement in lowering heart rate in patients with essential hypertension, which may contribute to the decreased risk of cardiovascular disease.

### 4.1. General Data

A total of 63 patients with hypertension were recruited for this study, and the average age was 40 years (DF group (40.09 ± 9.32) years, control group (40.80 ± 10.94) years). A total of 78.8% of the participants in the DF group were males, compared with 53.3% in control group, which is consistent with the investigation report of hypertension epidemiology in China [2]. The average BMI of participants in the DF group and control group were (24.93 ± 3.12) kg/m^2^ and (25.38 ± 4.83) kg/m^2^, respectively, indicating that most participants were overweight, which may be related to no exercise or low exercise intensity. Rong et al. [38] pointed out that overweight is a common risk factor for hypertension, which is consistent with our study. While the duration of hypertension was about 1.6 years, 96.7% of patients had no complications, up to 33.3% of patients took anti-hypertensive drugs, and most patients had good blood pressure control. Smoking and drinking are risk factors for the occurrence of hypertension [39], and most participants kept the habit of non-smoking and non-drinking, which may have been due to the fact that young patients who are more educated have a better ability to learn and can change their behavior, which may worsen the disease process [40].

### 4.2. Compliance to Dietary Fiber Intake

The recommended quantity of dietary fiber by the World Health Organization is (25~35) g/d [35]. However, with increasing economic development, the increased consumption of energy-dense and processed foods has led to a reduced consumption of dietary fiber. The quantities of dietary fiber intake in previous studies were far below the recommended doses. Liu et al. [41] conducted an investigation that showed that the average daily dietary fiber intake of HTN patients was 11.25 ± 4.88 g, which is consistent with the study by Qi et al. [42], who found that the average daily DF intake was 10.17 ± 4.38 g in HTN patients. In this study, HTN patients were provided with oat bran supplements, ensuring a daily consumption of DF of 21.45 ± 3.86 g/d. In addition, we found that there was an average of 6.2 bags/w of oat consumption. On this basis, we analyzed the effect of dietary fiber supplement on heart rate lowering.

### 4.3. Dietary Fiber on Heart Rate

Compared with office heart rate, ambulatory heart rate can provide more valuable information for the clinic, including circadian variation of heart rate, heart rate variability, nocturnal heart rate, and avoidance of white-coat effect [43]. In this study, we used ambulatory heart rate as the primary outcomes of study and analyzed the effect of dietary fiber on lowering 24 h average, maximum, and minimum heart rates. In addition, we explored the impact of dietary fiber on heart rate during daytime and nocturnal heart rate.

#### 4.3.1. Dietary Fiber on 24 h Heart Rates

24h maxHR is an important reference when determining maximal work capacity and maximal oxygen consumption [5], while 24h aveHR is an independent and effective predictor of increased risk of ischemic events, sudden death, and cardiovascular death in patients with cardiovascular disease [44]. Kolloch et al. [45] reported that the primary end point events, including death, myocardial infarction, or stroke, increased by 6% for each 5 beats/min increase in heart rate. Similarly, Lonn et al. [46] indicated that for every 10 beats/min increase in heart rate, major cardiovascular events, cardiovascular mortality, and all-cause mortality increased by 12%, 18%, and 15%, respectively. In this study, we found that after oat supplementation for three months, the 24h maxHR slowed down by 9.79 ± 18.96 bpm, while it was only 1.50 ± 12.88 bpm in the control group (*t* = −2.009, *p* = 0.049). At the same time, the result of within-group comparisons in the intervention group showed that after intervention, 24h aveHR decreased by 4.12 bpm (*p* = 0.022). While we did not find similar differences in the control group, the possible mechanisms might be that dietary fiber can modulate human gut microbiota and result in an increased short chain fatty acid (SCFA)-producing bacteria, which was confirmed by our previous study [23]. Xue et al. [23] conducted a randomized control trial using oat supplementation for three months in patients with hypertension, and the results indicated that Bifidobacterium (*p* = 0.019) and Spirillum (*p* = 0.006) were significantly increased. SCFAs include several carboxylic acids such as butyric acid (BA) produced by bacterial fermentation of dietary fibers [47]. Increased SCFA may influence the circulatory and the nervous system functions [48]. A pre-clinical study found that BA administered into the colon of rats produced prolonged decrease in HR [47]. An increase in the concentration of BA in the colon produces a significant hypotensive effect that depends on the afferent colonic vagus nerve signaling and GPR41/43 receptors, inhibiting the release of renin and thus playing a crucial role in regulation of HR [49,50]. Vagal afferents also express receptors that can sense SCFAs, providing another pathway for the HR-modulating effects of SCFAs [50]. In addition, SCFAs, in particular butyrate, have anti-inflammatory effects that are presumed to be mediated by inhibition of histone deacetylase (HDAC), which may decrease HR [51].

#### 4.3.2. Dietary Fiber on Daytime and Nighttime Heart Rates

Alterations in heart rate in individuals showed circadian variations, manifested by a relatively fast heart rate during the day and a slow heart rate during the night, which are mainly regulated by the neurohumoral system [52]. Gut microbiota also has circadian rhythmicity and can mediate the effects of disruption of circadian rhythms on human health [53,54]. Furthermore, dietary fiber can modulate gut microbiota and thus may further adjust variation of heart rate contributed to by circadian rhythms. In this study, after three months of oat bran intervention, D-aveHR, D-minHR, and D-max HR in the DF group decreased by 5.00 bpm, 3.64 bpm, and 8.97 bpm, respectively, while nighttime HR showed an insignificant decrease. The reason might be that autonomic nerves are hyperexcitable in the daytime [52], which may increase the expression of receptors such as GPR41/43, and the increased SCFA by dietary fiber supplement may interact with them, and thus the daytime heart rate fall was more pronounced. However, at night, autonomic nerves are in a hypotensive state [52], which may result in limited effect of dietary fiber. In addition, SCFA production is relatively less due to the gut microbiota’s circadian rhythmicity, so the heart rate-lowering effect of DF is not obvious.

### 4.4. Dietary Fiber on Medication Change

The first choice for drug treatment of hypertensive patients with increased heart rate is beta-blockers, which can slow down heart rate and reduce sympathetic nerve excitability [8]. Another common anti-hypertensive drug for hypertension and increased heart rate is calcium channel blockers, with negative muscle strength, negative conduction, and negative frequency, which can slow down the heart rate without inhibiting sympathetic nerves [8]. Moreover, there is ivabradine, which slows the heart rate and basically has no effect on blood pressure [8].

The results of this study showed that no participants in either group used ivabradine. Only 30.3% and 46.7% of participants in the DF group and control group used beta-blockers and calcium channel blockers, respectively. On one hand, with the short duration of hypertension, participants tended to control their blood pressure and heart rate with lifestyle changes rather than drug treatment, which is consistent with ESC/ESH guidelines for the management of hypertension. On the other hand, calcium channel blockers were indicated for elderly patients with hypertension [25], while the average age in this study was around 40 years.

## 5. Conclusions

On the basis of the results of our study, we conclude that dietary fiber (oat bran) supplementation might be beneficial in lowering HR in patients with essential hypertension and may be a useful strategy of effectively improving HR in patients with HTN.

## 6. Limitations

This study has some limitations. Firstly, the observational period was not long enough to assess the long-term outcomes that may have resulted from dietary fiber supplementation. Therefore, further long-term clinical trials are needed to verify the relationship between dietary fiber supplementation and heart rate. Secondly, our study addressed the association between dietary fiber intake and heart rate, and did not discuss the influence of different dietary fiber doses on heart rate. Therefore, more studies are needed in the future to address this aspect. Finally, further analysis of the characteristics of hypertensive people who do not respond to DF in lowering heart rate and finding the influencing factors is recommended.

## Figures and Tables

**Figure 1 nutrients-14-03148-f001:**
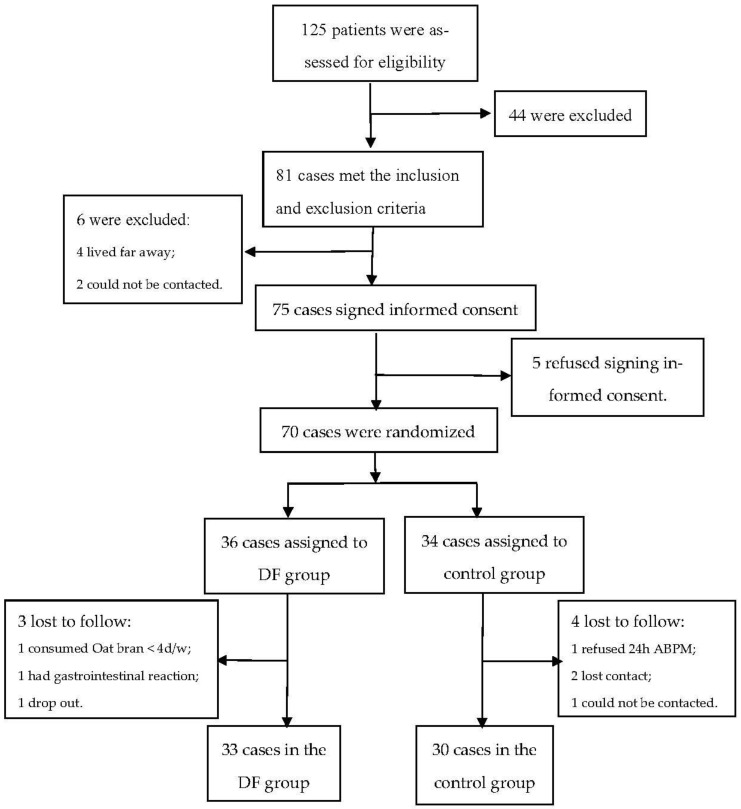
Trial flowchart. ABPM: ambulatory blood pressure monitoring; DF: dietary fiber.

**Table 1 nutrients-14-03148-t001:** Socio-demographic and clinical characteristics.

Characteristic		DF Group (*n* = 33)	Control Group (*n* = 30)	*t*/χ^2^/z	*p*
		x ± s/*n*(%)/ M(P25, P75)	x ± s/*n*(%)/ M(P25, P75)		
Ages (years)		40.09 ± 9.32	40.80 ± 10.94	0.736 ^a^	0.465
Sex	Male	26 (78.8)	16 (53.3)	4.582 ^b^	0.032
	Female	7 (21.2)	14 (46.7)		
Marital status	Unmarried	11 (33.3)	5 (16.7)	2.304 ^b^	0.129
	Married	22 (66.7)	25 (83.3)		
Education	Primary school	1 (3.0)	0 (0.0)	1.443 ^b^	0.695
	Junior middle school	4 (12.1)	5 (16.7)		
	High school/SSS	5 (15.2)	6 (20.0)		
	College or higher	23 (66.7)	19 (63.3)		
Professional situation	On the job	31 (94.0)	25 (83.3)	1.838 ^b^	0.399
	Retired	1 (3.0)	3 (10.0)		
	Job-waiting	1 (3.0)	2 (6.7)		
Medical payment	Self-funded	3 (9.0)	1 (3.3)	3.938 ^c^	0.140
	NRCMI	3 (9.0)	0(0.0)		
	Medical insurance	27 (81.8)	29 (96.7)		
Exercise	Yes	11 (33.3)	12 (40.0)	0.301 ^b^	0.583
	No	22 (66.7)	18 (60.0)		
DOS(hours/day)		6.48 (6.00, 7.63)	6.90 (6.00, 7.50)	−0.561 ^d^	0.743
Sleep quality	Poor	5 (15.2)	0(0.0)	5.491 ^b^	0.064
	Fair	17 (51.5)	21 (70.0)		
	Good	11 (33.3)	9 (30.0)		
Smoke	Yes	8 (24.2)	7 (23.3)	0.007 ^b^	0.933
	No	25 (75.8)	23 (76.7)		
Alcohol intake	Yes	9 (27.3)	9 (30.0)	0.057 ^b^	0.811
	No	24 (72.7)	21 (70.0)		
BMI (kg/m^2^)		24.93 ± 3.12	25.38 ± 4.83	0.436 ^a^	0.664
WHR		0.92 ± 0.18	0.96 ± 0.37	0.561 ^d^	0.743
Constipation	Yes	2 (6.1)	3 (10.0)	0.334 ^c^	0.563
	No	31 (93.9)	27 (90.0)		
DOH (years)		1.98 ± 3.70	1.67 ± 3.17	−0.182 ^a^	0.856
Anti-hypertensive drugs	Yes	6 (18.2)	10 (33.3)	1.904 ^c^	0.168
	No	27 (81.8)	20 (66.7)		
Comorbidity	Yes	2 (6.1)	5 (16.7)	1.790 ^c^	0.181
	No	31 (93.9)	25 (83.3)		
HR-lowing drugs	Yes	1	3	1.284 ^c^	0.257
	No	32	27		
SBP (mmHg)		131.30 ± 4.80	137.13 ± 8.09	3.514 ^a^	0.001 *
DBP (mmHg)		84.58 ± 4.72	85.40 ± 5.26	0.655 ^a^	0.515
Complication	Yes	0 (0.0)	1 (3.3)	1.118 ^c^	0.290
	No	33 (100.0)	29 (96.7)		

DF: dietary fiber; SSS: secondary specialized school; NRCMI: new rural cooperative medical insurance; DOS: duration of sleep; BMI: body mass index; WHR: waist–hip ratio; DOH: duration of hypertension; HR: heart rate; SBP: systolic blood pressure; DBP: diastolic blood pressure. ^a^ Independent samples *t*-test; ^b^ Pearson chi-squared test; ^c^ Fisher’s exact test; ^d^ Mann–Whitney U test; * *p* < 0.05.

**Table 2 nutrients-14-03148-t002:** Comparison of qualities of dietary nutrition (except DF) intake between two groups at baseline.

Characteristic	DF Group (*n* = 33)	Control Group (*n* = 30)	*t*	*p*
	x ± s/*n*(%)	x ± s/*n*(%)		
Total calories (kcal/day)	1660.55 ± 386.66	1607.41 ± 381.21	−0.543	0.589
Carbohydrate (g/day)	210.66 ± 66.34	202.17 ± 62.18	−0.517	0.607
Protein (g/day)	79.17 ± 22.97	77.38 ± 21.74	−0.315	0.754
Fat (g/day)	57.66 ± 22.83	54.66 ± 21.49	−0.530	0.598
Cholesterol (mg/day)	534.67 ± 290.31	510.48 ± 291.59	−0.327	0.745
Calcium (mg/day)	563.45 ± 172.85	577.57 ± 261.02	0.254	0.800
Potassium (mg/day)	1971.68 ± 467.21	1993.56 ± 663.66	0.153	0.879
Sodium (mg/day)	1282.50 ± 800.98	1391.69 ± 878.15	0.514	0.609
Vitamin D (μg/day)	3.10 ± 3.74	4.20 ± 5.20	0.962	0.340
Folic acid (μg/day)	118.32 ± 61.78	124.0 ± 96.51	0.284	0.777
Vitamin B6 (μ/day)	0.36 ± 0.37	0.24 ± 0.18	−1.567	0.122
Magnesium (mg/day)	296.65 ± 77.49	310.45 ± 115.51	0.558	0.579
Zinc (mg/day)	10.18 ± 2.59	11.47 ± 4.51	1.402	0.166
Selenium (μg/day)	52.47 ± 17.47	55.51 ± 27.11	0.215	0.830

DF: dietary fiber.

**Table 3 nutrients-14-03148-t003:** Comparison of the quantity of DF intake between the two groups.

DF (g/d)	DF Group (*n* = 33)	Control Group (*n* = 30)	*t*	*p*
	x ± s/*n*(%)	x ± s/*n*(%)		
Baseline	11.07 ± 3.73	10.66 ± 4.27	−0.388	0.700
3 month	21.45 ± 3.86	10.40 ± 3.76	−10.75	<0.001 *

DF: dietary fiber; * *p* < 0.05.

**Table 4 nutrients-14-03148-t004:** Comparison of HR between/within two groups.

HR	Time	DF Group (*n* = 33) $\bar{x}$ ± s	Control Group (*n* = 30) $\bar{x}$ ± s	*t*/F	*p*
24h aveHR	Baseline	75.15 ± 9.69	73.73 ± 8.21	−0.624	0.535
	3 month	71.03 ± 9.91	73.00 ± 7.62	0.734	0.466
	*t*	2.401	0.629		
	*p*	0.022 *	0.534		
	3 m, adjusted	70.74 ± 1.59	72.70 ± 1.80	0.924	0.341
	MD	4.12 ± 9.86	0.73 ± 6.39	−1.600	0.115
24hminHR	Baseline	54.15 ± 6.51	55.33 ± 7.55	0.667	0.507
	3 month	53.88 ± 7.65	54.63 ± 5.39	0.336	0.738
	*t*	0.284	0.639		
	*p*	0.778	0.528		
	3 month, adjusted	53.81 ± 1.20	54.47 ± 1.31	0.197	0.684
	MD	0.27 ± 5.51	0.700 ± 6.00	0.295	0.769
24h maxHR	Baseline	103.03 ± 18.43	99.67 ± 13.74	−0.822	0.414
	3 month	93.24 ± 15.26	98.17 ± 16.99	1.099	0.276
	*t*	2.966	0.638		
	*p*	0.006 *	0.529		
	3 month, adjusted	92.96 ± 2.93	96.80 ± 3.27	1.062	0.308
	MD	9.79 ± 18.96	1.50 ± 12.88	−2.009	0.049 *
D-aveHR	Baseline	78.30 ± 10.53	76.43 ± 8.30	−0.777	0.440
	3 month	73.30 ± 10.13	74.97 ± 8.08	0.637	0.527
	*t*	2.956	1.239		
	*p*	0.006 *	0.225		
	3 month, adjusted	73.17 ± 1.66	74.54 ± 1.93	0.447	0.507
	MD	5.00 ± 9.71	1.46 ± 6.48	−1.681	0.098
D-minHR	Baseline	60.61 ± 9.05	60.17 ± 8.01	−0.203	0.840
	3 month	56.97 ± 8.01	57.97 ± 7.37	0.490	0.626
	*t*	2.048	1.760		
	*p*	0.049 *	0.089		
	3 month, adjusted	56.97 ± 1.45	57.82 ± 1.49	0.159	0.692
	MD	3.64 ± 10.20	2.14 ± 6.95	−0.649	0.518
D-maxHR	Baseline	101.76 ± 20.73	98.83 ± 12.31	−0.672	0.504
	3 month	92.79 ± 15.26	98.17 ± 16.39	1.213	0.230
	*t*	2.278	0.264		
	*p*	0.030 *	0.794		
	3 month, adjusted	92.60 ± 2.96	96.72 ± 3.31	1.169	0.285
	MD	8.97 ± 22.63	0.67 ± 13.86	−1.735	0.088
N-aveHR	Baseline	61.64 ± 7.45	63.53 ± 8.90	0.920	0.361
	3 month	62.91 ± 9.48	63.20 ± 6.14	0.219	0.828
	*t*	−0.868	0.241		
	*p*	0.392	−0.668		
	3 month, adjusted	62.87 ± 1.37	63.11 ± 1.54	0.032	0.859
	MD	−1.27 ± 8.43	0.33 ± 7.57	0.793	0.431
N-minHR	Baseline	53.94 ± 6.71	56. 83 ± 9.20	1.436	0.156
	3 month	54.88 ± 7.96	54.80 ± 6.05	−0.024	0.981
	*t*	−0.866	1.698		
	*p*	0.392	0.100		
	3 month, adjusted	54.54 ± 1.25	55.06 ± 1.40	0.072	0.790
	MD	−0.94 ± 6.23	2.03 ± 6.56	1.844	0.070
N-maxHR	Baseline	73.58 ± 12.20	74.17 ± 11.91	0.194	0.847
	3 month	74.97 ± 14.88	73.97 ± 10.14	−0.257	0.798
	*t*	−0.493	0.097		
	*p*	0.625	0.923		
	3 month, adjusted	75.08 ± 2.26	73.83 ± 2.52	0.108	0.744
	MD	−1.39 ± 16.23	0.200 ± 11.26	0.449	0.655

24h aveHR: 24 h average heart rate; 24hminHR: 24 h minimum heart rate; 24h maxHR: 24 h maximum heart rate; D-aveHR: aveHR during day time; D-minHR: minHR during day time; D-manHR: maxHR during day time; N-aveHR: aveHR during the night; N-minHR: minHR during the night; N-maxHR: maxHR during the night. MD: mean difference. Adjusted for age, sex, marital status, medical payment, sleep quality, anti-hypertensive drugs, comorbidity, SBP, DBP, vitamin B6, and zinc; * *p* < 0.05.

**Table 5 nutrients-14-03148-t005:** Intention-to-treat analysis of comparison of HR between/within two groups.

HR	Time	DF Group (*n* = 36) $\bar{x}$ ± s	Control Group (*n* = 34) $\bar{x}$ ± s	*t*	*p*
24h aveHR	Baseline	74.97 ± 9.37	74.18 ± 7.99	−0.381	0.704
	3 month	71.19 ± 9.58	73.53 ± 7.49	0.1.31	0.262
	*t*	2.386	0.629		
	*p*	0.023 *	0.533		
	MD	3.77 ± 9.50	0.65 ± 5.99	−1.638	0.106
24hminHR	Baseline	54.36 ± 6.47	55.47 ± 7.12	0.683	0.497
	3 month	54.11 ± 7.53	54.85 ± 5.14	0.479	0.634
	*t*	0.285	0.640		
	*p*	0.778	0.527		
	MD	0.25 ± 5.27	0.62 ± 5.63	0.282	0.779
24h maxHR	Baseline	102.64 ± 17.67	99.91 ± 13.06	−0.731	0.467
	3 month	93.67 ± 14.67	98.59 ± 16.36	1.326	0.189
	*t*	2.936	0.639		
	*p*	0.006 *	0.527		
	MD	8.97 ± 18.34	1.32 ± 16.08	−2.048	0.044 *
D-aveHR	Baseline	78.08 ± 10.16	76.85 ± 8.08	−0.559	0.578
	3 month	73.50 ± 9.77	75.56 ± 7.97	0.963	0.339
	*t*	2.929	1.237		
	*p*	0.006 *	0.225		
	MD	4.58 ± 9.38	1.29 ± 6.10	−1.717	0.089
D-minHR	Baseline	60.61 ± 8.79	60.26 ± 7.63	−0.176	0.861
	3 month	57.28 ± 7.89	58.32 ± 7.09	0.582	0.562
	*t*	2.039	1.753		
	*p*	0.049 *	0.089		
	MD	3.33 ± 9.81	1.82 ± 6.45	−0.697	0.488
D-maxHR	Baseline	101.47 ± 20.73	99.18 ± 12.12	−0.580	0.564
	3 month	93.25 ± 14.68	98.59 ± 16.36	1.438	0.155
	*t*	2.265	0.264		
	*p*	0.030 *	0.793		
	MD	8.22 ± 21.78	0.59 ± 12.99	−1.768	0.082
N-aveHR	Baseline	61.75 ± 7.38	64.15 ± 8.95	1.226	0.225
	3 month	62.92 ± 9.26	63.85 ± 6.63	0.484	0.630
	*t*	−0.868	0.242		
	*p*	0.391	0.810		
	MD	−1.16 ± 8.07	0.29 ± 7.09	0.803	0.425
N-minHR	Baseline	54.17± 6.66	58.09 ± 11.61	1.745	0.086
	3 month	55.03 ± 7.80	56.29 ± 9.90	0.596	0.553
	*t*	−0.866	1.639		
	*p*	0.392	0.100		
	MD	−0.86 ± 5.97	1.79 ± 6.18	1828	0.072
N-maxHR	Baseline	73.44 ± 11.94	73.74 ± 11.69	0.103	0.918
	3 month	74.72 ± 14.47	73.56 ± 10.12	−0.388	0.700
	*t*	−0.494	0.098		
	*p*	0.624	0.923		
	MD	−1.28 ± 15.52	−0.38 ± 10.02	0.456	0.650

24h aveHR: 24h average heart rate; 24hminHR: 24h minimum heart rate; 24h maxHR: 24h maximum heart rate; D-aveHR: 24h aveHR during day time; D-minHR: minHR during day time; D-maxHR: maxHR during day time; N-aveHR: aveHR during the night; N-minHR: minHR during the night; N-maxHR: maxHR during the night. MD: mean difference; * *p* < 0.05.

**Table 6 nutrients-14-03148-t006:** Comparison of the HR-lowing drug (β-receptor blocker/calcium ion antagonist) at 3 m between two groups.

Changes of Medication Dosage	DF Group (*n* = 33)	Control Group (*n* = 30)	χ^2^	*p*
Reduced	4 (12.1)	2 (6.7)	5.926	0.115
No change	3 (9.1)	7 (23.3)		
Increased	3 (9.1)	5 (16.7)		
Not taking medicine	23 (69.7)	16 (53.3)		

HR: heart rate.

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
