# Peer review of "Effect of Dietary Fiber (Oat Bran) Supplement in Heart Rate Lowering in Patients with Hypertension: A Randomized DASH-Diet-Controlled Clinical Trial"

_nutrients, 2022, doi:10.3390/nu14153148_

Round 1
Reviewer 1 Report
I carefully read the re-submitted and revised version of the manuscript by Ju et al., that is not significantly improved in comparison with the previous one. In effect, the authors have ignored most of my comments and suggestions, which I propose again below
- English language needs to be extensively improved.
- The authors should consider to refer to doi: 10.3390/nu12030686, doi: 10.33963/KP.15468 and doi: 10.1007/s40292-021-00474-6 in their manuscript.
- The authors should specify if they have performed the Levene's test before the Student's T test.
- The authors should specify if they used a validated score to assess presence of constipation.
- The authors should specify how the treatment compliance was assessed.
- Lines 98-99: The authors claim that their study was single blinded. However, a diet-controlled study cannot be a single-blinded study, since there is no control treatment as per study's design.
- Line 108: The authors refer to "interventions". However, they should more properly refer to "intervention".
- The authors administered to controls DASH diet. Then, the authors should clearly claim (also in the title of the manuscript) that their study was a not blinded diet-controlled clinical study.
Author Response
Responses to the reviewer 1:
We would like to express our sincere thanks to the reviewers for the constructive
and positive comments. The comments have been carefully taken into account and a new revised submission has been uploaded. We highlighted all the altered passages in light RED in the revised manuscript. The page number of revised manuscript was
used in the followed responses. The responses are as follows.
- English language needs to be extensively improved.
Response to Reviewer comment No. 1: According to your suggestion, we have made an extensive English revisions for this manuscript.
- The authors should consider to refer to doi: 10.3390/nu12030686, doi: 10.33963/KP.15468 and doi: 10.1007/s40292-021-00474-6 in their manuscript.
Response to Reviewer comment No. 2: We have learned the contents of references you provided and thought it is important for our study and the manuscript. Thus we referred two of them as the references in the manuscript. Please see No. 14 and 27 of references. Thank you!
- The authors should specify if they have performed the Levene's test before the Student's T test.
Response to Reviewer comment No. 3: We had performed the Levene's test before the Student's T test, and we have added it in the part of statistical analysis.
- The authors should specify if they used a validated score to assess presence of constipation.
Response to Reviewer comment No. 4: We used the validated Rome III criteria to assess constipation .Criteria are as follows: straining (≥25%); lumpy or hard stools (≥25%); sensation of incomplete evacuation (≥25%); sensation of anorectal obstruction/blockade (≥25%); manual maneuvers to facilitate defecations (≥25%) and < 3 times/w of defecations. Patients who met at least 2 of the 6 criteria were diagnosed as constipation. We have added it in the methodology of manuscript.
- The authors should specify how the treatment compliance was assessed.
Response to Reviewer comment No. 5: Sorry for our unclear expression about compliance to oat bran consumption. We have clarified the evaluating method of the compliance to oat bran consumption: based on the collected number of bags of oat bran consumed every week. The content was showed in the methodology of manuscript.
- Lines 98-99: The authors claim that their study was single blinded. However, a diet-controlled study cannot be a single-blinded study, since there is no control treatment as per study's design.
Response to Reviewer comment No. 6: Sorry for our mistake. We have modified it in the methodology of the manuscript.
- Line 108: The authors refer to "interventions". However, they should more properly refer to "intervention".
Response to Reviewer comment No. 7:: Sorry for our mistake. We have modified “interventions” to “intervention” in line 112.
- The authors administered to controls DASH diet. Then, the authors should clearly claim (also in the title of the manuscript) that their study was a not blinded diet-controlled clinical study.
Response to Reviewer comment No. 8: Thanks for your suggestion. We modified “a randomized controlled trial” to “a randomize DASH diet-controlled trial” in the title and the manuscript.

Reviewer 2 Report
Methods could be improved (blinding, sample size, duration of follow up).
There isn't a quantitative evaluation of sleep quality (for example Pittsburgh, Epworth, AHI).
Results seems to be original findings, but don't provide a significative contribution to scientific research on hypertension cause of major concerns about methods.
Author Response
Responses to the reviewer 2:
We would like to express our sincere thanks to the reviewers for the constructive
and positive comments. The comments have been carefully taken into account and a
new revised submission has been uploaded. We highlighted all the altered passages in
light RED in the revised manuscript. The page number of revised manuscript was
used in the followed responses. The responses are as follows.
Point 1: Methods could be improved (blinding, sample size, duration of follow up).
Response to Reviewer comment No. 1: In the corresponding sections, the method of sample size (line 131-139) and duration of follow up (line 157) have been changed. As a diet-controlled study cannot be a single-blinded study, so we modified it in the manuscript and no blinding involved.
Point 2:There isn't a quantitative evaluation of sleep quality (for example Pittsburgh, Epworth, AHI).
Response to Reviewer comment No. 1: We used a Visual Analog Rating Scale as the tool to evaluated the quality of sleep which was added the methodology in line 203-205.
Point 3: Results seems to be original findings, but don't provide a significative contribution to scientific research on hypertension cause of major concerns about methods.
Response to Reviewer comment No. 1: We have added some significative contribution of HR lowering in the discussion (line 438-442).

Round 2
Reviewer 1 Report
Dear Editor,
I carefully read the further revised version of the manuscript that is improved in comparison with the previous version.
This manuscript is a resubmission of an earlier submission. The following is a list of the peer review reports and author responses from that submission.
Round 1
Reviewer 1 Report
The Authors address a few limitation to their very limited experiment. The first one is that the quantity of DF intake following supplemention did not reach the minimum dose (25g/d) recommended by the guidelines...." and point with no proof that this "..... may explain the fact that the significant difference in heart rate between the two groups was only found in the 24hmaxHR....." so that "... In the future studies, it may be essential to use more DF (target dose), in order to observe the changes in heart rate in patients with hypertension....". I am astonished with these comments, inasmuch as they do not realize that in both intention to treat analysis and overall assessment there were (Tables 4 and 5) only very borderline significant differences between groups in 24-hours HR which certainly points to the low power of the study which was not provisionally calculated. And all this in front of a large didfference in gender distribution that may actually account for most of the observed differential changes.
The study should thus be repeated with the actually recommended quantity of DF and making gender-moderated power calculations to enable sensitive conclusions.
Reviewer 2 Report
Dear Editor,
I carefully read the manuscript by Ju et al.
My comments and suggestions for the authors are the following:
- English language needs to be extensively improved. In particular, citing almost unknown authors in the manuscript makes the reading difficult.
- The manuscript is not balanced in its parts. In particular, the background paragraph should be shortened.
- The authors should consider to refer to doi: 10.3390/nu12030686, doi: 10.33963/KP.15468 and doi: 10.1007/s40292-021-00474-6 in their manuscript.
- The authors should specify if they have performed the Levene's test before the Student's T test.
- The authors should replace "gender" with "sex" throughout the manuscript. As a matter of fact, sex refers to “the different biological and physiological characteristics of males and females, such as reproductive organs, chromosomes, hormones, etc.” Gender refers to "the socially constructed characteristics of women and men – such as norms, roles and relationships of and between groups of women and men.
- The authors should specify if they used a validated score to assess presence of constipation.
- The limitations of the study should be further and more comprehensively discussed.
- The authors should specify how the treatment compliance was assessed.
- The authors should better describe the control treatment.